# Biochemical Properties of Black and Green Teas and Their Insoluble Residues as Natural Dietary Additives to Optimize In Vitro Rumen Degradability and Fermentation but Reduce Methane in Sheep

**DOI:** 10.3390/ani12030305

**Published:** 2022-01-26

**Authors:** Diky Ramdani, Anuraga Jayanegara, Abdul Shakoor Chaudhry

**Affiliations:** 1Department of Animal Production, Faculty of Animal Husbandry, Universitas Padjadjaran, Sumedang 45363, Indonesia; 2Department of Animal Feed Science and Technology, Faculty of Animal Science, IPB University, Bogor 16680, Indonesia; anuragaja@apps.ipb.ac.id; 3School of Natural and Environmental Sciences, Newcastle University, Newcastle upon Tyne NE1 7RU, UK; abdul.chaudhry@newcastle.ac.uk

**Keywords:** black tea, green tea, spent tea leaves, dietary additives, ruminants

## Abstract

**Simple Summary:**

Animal nutritionists are challenged to increase animal production with respect to competitiveness and efficiency yet at the same time produce products that are healthy for the consumer and friendly to the environment. Black (BTL) or green (GTL) tea leaves and their spent tea leaf residues (STL) are rich in polyphenols, and can be used to not only supplement animal diets but also reduce the environmental burden of their safe disposal. These tea products were assessed for their potential use as natural dietary additives in ruminant diets to not only optimize rumen in vitro digestibility and fermentation but also reduce methane (CH_4_) emission. Both types of tea leaves were effective in significantly reducing ammonia (NH_3_) and CH_4_ production without decreasing rumen degradability. Tea leaves and their STL inclusions increased the acetate to propionate (A:P) ratio significantly. The GTL-containing diets had significantly greater rumen degradability but lower rumen NH_3_ compared with the BTL diets. Decreased rumen NH_3_ production in ruminants can be a possible sign of increasing bypass protein whilst reduced rumen CH_4_ release can be useful for enhancing energy use efficiency and environmentally friendly ruminant production.

**Abstract:**

Black (BTL) or green (GTL) tea and their spent tea (STL) leaves can be used as natural dietary additives for ruminants. Experiment 1 used a 3 × 2 × 2 factorial arrangement, with four replicates (*n* = 4) to test the effects of three different inclusions of tea leaves at 0 (control), 50, and 100 g/kg DM of two different tea types (BTL and GTL) in two different total mixed diets containing either ryegrass hay (RH) or rice straw (RS) on in vitro rumen organic matter degradability (IVOMD), volatile fatty acids (VFA), pH, ammonia (NH_3_), and methane (CH_4_) outputs over a 24 h incubation time. Experiment 2 followed a 3 × 2 × 2 factorial arrangement, with eight replicates (*n* = 8) to study the impacts of three different STL inclusions at 0, 100, and 200 g/kg DM of two different STL types (black and green) into two different total mixed diets containing either RH or RS on the same in vitro measurements. Both types of tea leaves decreased NH_3_ (*p* < 0.001) and CH_4_ (*p* < 0.01) without affecting (*p* > 0.05) rumen degradability, but the effect of their STL was less remarkable. Tea leaves and their STL inclusions improved (*p* < 0.01 and *p* < 0.001, respectively) the acetate to propionate (A:P) ratio. Compared with BTL, GTL containing diets had higher IVOMD (*p* < 0.05) and A:P ratio (*p* < 0.05) but lower NH_3_ (*p* < 0.001). Reduced rumen NH_3_ and CH_4_ outputs can be useful for protein and energy use efficiency while an increased A:P ratio might lead to increased milk fat synthesis and reduced low-fat milk syndrome. The surplus or wasted tea leaf products could be used as sustainable sources of nutrients to optimize rumen function and minimize environmental impacts of feeding ruminant animals.

## 1. Introduction

Black (BTL) or green (GTL) tea leaves and their spent (STL) residues contain considerable amounts of protein, minerals, and plant secondary metabolites including tannins and saponins [1]. Catechins and theaflavins [2,3] are the major polyphenols in GTL and BTL, respectively. Both tea leaves also contain considerable amounts of alkaloids, mainly caffeine [3]. Tannins are able to bind and protect feed proteins in the ruminant diets. This protection increases available rumen bypass protein and non-ammonia nitrogen (non-NH_3_ N) supplies to be further absorbed in the small intestine [4]. Since NH_3_ is used as one of the main sources of N for rumen microbiota, its quick production may exceed the capability of microbes to use it, especially if it is not synchronized by supplying additional soluble carbohydrates in diets. This asynchrony can result in more NH_3_ supply than is required, which after being absorbed through the rumen wall, flows into the blood stream and liver, or finally excreted as N waste in the urine [5]. A previous study [6] concluded that tea tannin inclusion in the diet of fattening lambs could increase their weight gains without affecting feed intakes and protein digestibility. This is an indication that bound protein was not well digested in the rumen but was available as bypass protein that is able to be absorbed in the small intestine.

Dietary tannins can decrease rumen methane (CH_4_) output [4,7,8]. Equally, tea saponins can decrease rumen CH_4_ yield by decreasing protozoa and the methanogenic activity of relevant bacteria [9,10,11]. Methane yield in the rumen is linked to a loss of dietary gross energy by 2–14% [12,13,14]. Livestock sectors, predominantly ruminants via their eructation and manures, contribute up to 40% of total CH_4_ production by agricultural activities [15,16,17]. Therefore, CH_4_ reduction in ruminant animals is an intention, not only for environmental benefit, but also for nutrient use efficiency. Nonetheless, reduced CH_4_ and NH_3_ productions in the rumen due to significant decreases in protozoa and bacterial populations may affect the optimal rumen function leading to poor feed degradability. The challenge is therefore to reduce CH_4_ and NH_3_ productions (for increasing potential bypass protein and reducing N waste) without affecting rumen function and feed degradability. If the bioactive constituents of tea leaf products can do so, these products could be exploited as natural alternatives to substitute growth-promoters or antibiotics that have been prohibited in the European countries since 2003 (1831/2003; EC, 2003). This ban has been expanded to Indonesia since 2017 (Indonesian Ministry of Agriculture Regulation No. 14/2017).

In humans, tea catechins [18,19], theaflavins [20,21], and caffeine [22,23] from tea leaves were reported to have antioxidant and cancer prevention properties. However, their potential advantage as additives in ruminant diets is not well-understood yet. Unlike original tea leaves, STL has been studied in vitro [24,25] and in vivo [26,27] for their suitability as ruminant feedstuffs but none of previous studies have looked at their potential to mitigate rumen CH_4_ output. Thus, this study examined the effects of tea leaves (BTL or GTL) and their STL as additives in ryegrass (RH) or rice straw (RS)-based diets on in vitro rumen degradability, fermentative profiles, and CH_4_ yield. Here, representative samples of RH and RS were used respectively as moderate and low-quality forages, that may be available to ruminants in different production situations worldwide.

## 2. Materials and Methods

The research was done as two separate in vitro rumen experiments by using the following arrangements. Here, experiment 1, with a quadruplicated 3 × 2 × 2 factorial arrangement, examined the effects of 3 different inclusions of tea leaves at 0 (control), 50, and 100 g/kg DM of 2 different tea types (black and green) in 2 different total mixed diets containing either RH or RS on in vitro organic matter degradability (IVOMD), volatile fatty acids (VFA), pH, NH_3_, and gas characteristics over 24 h incubation time. Experiment 2 followed a 3 × 2 × 2 factorial model, with 8 replicates to investigate the effects of 3 different STL inclusions at 0, 100, and 200 g/kg DM of 2 different STL types (black and green) in 2 different total mixed diets containing either RH or RS on the same in vitro measurements over 24 h incubation time as described above. Eight replicates in Experiment 2 consisted of 4 replicates of each STL that was processed in the laboratory at Newcastle University and 4 replicates of each STL that was collected from the company for both black and green STL as explained below.

Initially, the concept of adding either black or green STL in diets was tested at a higher level of up to 20% because each STL was recognized as the residual waste after brewing the original green or black tea leaves. However, the doses of original green or black tea leaves were reduced to 5 and 10% DM. Although high to moderate inclusion levels of tea leaves or STL were tested for the in vitro studies, the intention was to further reduce their dietary levels for the subsequent animal studies.

### 2.1. Samples of Tea Leaves, Dietary Ingredients, and Diet Preparation

Representative samples of BTL and GTL were received from a tea fabricating company (PT. Kabepe Chakra, West Java, Bandung Regency, Indonesia). Here, BTL and GTL were graded as Broken Orange Pekoe Fanning (Code: BOPF #355) and Sow Mee (Code: SM #315), respectively. Both BTL or GTL were sampled from three different respective batches which were then equally mixed and used to obtain relevant samples of BTL or GTL for Experiment 1 and to prepare the STL for Experiment 2 as previously described [1]. Briefly, the samples of STL were produced in the laboratory where 2.8 g of each black (SBTL) or green (SGTL) tea leaves was extracted in 300 mL of boiling water for 5 min. The extraction was repeated to get adequate samples of pooled STL for the entire study.

The company STL comprising either black (CSBTL) or green (CSGTL) leaves were collected from a tea beverage company (PT Coca-Cola Amatil, West Java, Bekasi City, Indonesia). The CSGTL and CSBTL samples representing the insoluble residues were regarded as the waste products after transferring the soluble liquids into ready-to-drink tea bottles branded as ‘frestea’. Each company STL was randomly sampled from different parts of freshly produced STL waste piles at different company sites. These samples were then pooled and oven-dried at 55 °C before being transported by air to Newcastle University along with the BTL and GTL samples.

Samples of RH and concentrate (CON, as one of the main dietary ingredients) were obtained from Cockle Park farm, Newcastle University during spring season whereas a dried form of RS (variety IR50) was collected from Bangladesh. The CON represented a typical complementary feed to prepare a forage containing total mixed diet for ruminants. It contained (g/kg DM) sugar beet pulp (260), soybean meal (220), maize distiller’s grain (150), mixture of barley and wheat (260), molasses (60), and a mineral mix (30, Scotmin Nutrition, Scotland, Ayr, UK). Before diet formulation and chemical analysis, each dietary sample was dried using an oven at 55 °C and ground using a sample mill (Cyclotec 1093, Tecator, Höganäs, Sweden) through 1 mm sieve. The formulae of the experimental diets containing either tea leaves or their STL are presented in Table 1.

### 2.2. Rumen Liquor Collection and Buffered Inoculum Preparation

The rumen liquor (RL) for the first in vitro trial was taken during summer (August) from 2 over 4-months-old freshly slaughtered Texel X Mule lambs that were fed grass-based diets throughout their post-weaning period. The RL for the second in vitro trial was collected during spring (June) from 2 over 3-months-old freshly slaughtered Cheviot lambs that were fed grass-based diets with cereal supplementation. All the RL were collected from a local abattoir (Linden Foods Ltd., Cramlington, UK). In each experiment, the pooled RL from the first set of lambs was used for replicates 1 and 2 while the pooled RL from the second set of lambs was used for replicates 3 and 4. After slaughtering, each rumen specimen was directly collected and cut open to obtain its semi-degraded feed contents. The rumen contents were then immediately filtered via two layers of a cheesecloth on a funnel to obtain RL into a pre-warmed insulated flask (Thermos Ltd., Leeds, UK). Each flask was closed tightly to maintain anaerobic conditions inside the flask. The flasks containing RL were then brought to the laboratory for their use within 1 h collection. Subsequently, measured amount of each RL was transferred into a pre-warmed dark bottle containing buffer solution [28] while kept in a water-bath (39 °C) to prepare buffered RL at 1:2 ratio of RL:buffer solution. The bottles comprising buffered RL were further purged with CO_2_ to remove O_2_ and closed tightly with a 50 mL dispenser (Fisher Scientific, Loughborough, UK). CO_2_ flushing was also done to adjust pH of each buffered inoculum to about 7 ± 0.2.

### 2.3. In Vitro Method and Sampling

The procedure of this in vitro study is referred to in the previous study [29]. Briefly, 200 ± 4 mg of each sample was placed inside a glass syringe (50 mL capacity, Camlab Ltd., Cambridge, UK), lubricated thinly with Vaseline, and fitted with a 4 way-male-slip stopcock (Cole Palmer Instrument, St Neots, UK). Each syringe was filled by 20 mL buffered inoculum before being closed and stored in a shaking water bath at 39 °C for incubation. Total gas production (tGP) in every syringe was quantified at 0, 2, 4, 6, 8, 18, 20, 22, and 24 h. After incubation, the warm water in the water bath was substituted with adequate ice to discontinue fermentation in the syringes. Approximately 10 mL gas from each incubated syringe was transferred into an additional empty syringe from where the gas was further transferred to a 12 mL evacuated gas tube (Labco Exetainer, Labco Ltd., Lampeter, UK) using an attached needle of the stopcock for CH_4_ and CO_2_ analyses. Moreover, whole contents in each syringe (inoculum and the residues) were transferred into a pre-weighted tube (polyethylene, 50 mL size) for the pH, NH_3_, VFA, and IVOMD analyses at 24 h of incubation time. The pH was measured by a calibrated pH meter (pH 309, Hanna Instruments Ltd., Leighton Buzzard, UK). All tubes were then centrifuged (Baird & Tatlock Ltd., London, UK) for 10 min at 2500 rpm to collect supernatants for VFA and NH_3_ analyses. The insoluble residues were then water-washed and dried at 80 °C for IVOMD quantification as explained in the previous study [30]. Two blank syringes representing buffered RF only were also proceeded alongside the samples in each experiment for their use to correct the degradability, tGP, and other fermentation profiles. 

### 2.4. VFA, NH_3_, CH_4_, and CO_2_ Analyses

The procedures of these VFA, NH_3_, and CH_4_ analyses were those that were previously described [29] by using a gas chromatograph (Shimadzu GC-2014, Kyoto, Japan), Pentra 400 (Horriba Ltd., Kyoto, Japan), and a gas chromatograph–mass spectroscope (Fisons 8060 GC, Milan, Italy), respectively. The CO_2_ analysis was done simultaneously with CH_4_ [29] using a curve calibration standard of pure CO_2_ (BOC Industrial Gases, Middlesbrough, UK).

### 2.5. Proximate, Detergent Fibre, and Secondary Metabolite Analyses

The protocols of the Association of Official Analytical Collaboration [31] were applied to analyze dry matter (DM, method 934.01), ash, organic matter (OM, method 942.05), and ether extract (EE, method 920.39). Total nitrogen (N) (N × 6.25 = Crude Protein, CP) was analyzed by Elementar Vario Macro Cube (Elementar, Hanau, Germany). The neutral detergent fibre (NDFom) contents were analyzed as previously described [30] without using amylase, sodium sulphite, and dekalin while acid detergent fibre (ADFom) and lignin (ADLom) were determined as previously reported [32]. The NDFom and ADFom contents were quantified without ash. Metabolizable energy (ME) was estimated by utilizing the equation of a previous study [33].

Total phenols (TP) and total tannins (TT) were examined by the Folin Ciocalteu procedure as explained previously [34] with tannic acid (Fisher Scientific, Loughborough, UK) as the standard reference. The procedure of [35] was applied for total saponin (TS) measurement by utilizing diosgenin (Molekula Ltd., Gillingham, UK) as a standard. A UV/VIS-spectrophotometer (Libra S12, Biochrom Ltd., Cambridge, UK) was employed in the TP, TT, and TS analyses. 

### 2.6. Mineral, Alkaloid and Polyphenol Analyses

Individual mineral, alkaloid, and polyphenol analyses were conducted for the tea leaf products only. The preparations of samples, standards, and extractions for mineral analysis on a Varian Vista-MPX CCD by simultaneous Inductively Coupled Plasma-Optical Emission Spectroscopy (ICP-OES) (Varian Inc., Mulgrave, Australia) were similar to the previous study [1]. The preparations of samples, standards, and extractions for simultaneous alkaloid and polyphenol analyses using high performance liquid chromatography (HPLC, Shimadzu, Kyoto, Japan) were similar to the previous studies [2,3].

### 2.7. Calculation and Statistical Analysis

Different data sets were organized in rows and columns in relevant spreadsheets of Microsoft Excel and carefully checked before their transfer to the respective data sheets in Minitab 17. The data sets were then tested for the Anderson−Darling normality test at *p* > 0.05, which suggested that all the data sets were normally distributed. An one-way analysis of variance (ANOVA) was utilized to compare chemical compositions of diet ingredients including tea leaves and their STL from triplicate analysis (*n* = 3) for statistical difference at *p* < 0.05. The Fit General Linear Model ANOVA was applied to analyze the statistical effects of inclusion levels, tea types, and diet types along with their interactions on in vitro degradability, fermentation profiles, and CH_4_ output in experiment 1 (*n* = 4) and to analyze the statistical effects of inclusion levels, STL types, and diet types on the same in vitro measurements in experiment 2 (*n* = 8). Tukey’s test was used to compare means for statistical difference at *p* < 0.05. The cumulative gas production values for tea leaves and their STL were plotted against time by using the scatter graph option in Excel. Appropriate trend lines alongside regression equations were then obtained by using the curvilinear option to find the relationship, as determined by R^2^, between cumulative gas production and incubation times.

## 3. Results

### 3.1. Chemical Properties

Table 2 shows mean chemical compositions of the experimental dietary ingredients. Black tea leaves had higher NDFom and ADFom content but less ADLom, EE, and secondary metabolites compared with GTL. Original BTL and GTL had higher DM and ash than their respective STL. Black tea leaves and GTL also had greater secondary metabolites such as TP, TT, and TS contents than their corresponding STL and other dietary ingredients such as RH, RS, and CON. All tea leaf products had higher CP than RH, RS, and CON. Ryegrass hay contained higher OM, EE, and ME contents but lower fiber fractions than RS. Table 3 shows the mean mineral components in different tea leaf products. Original BTL and GTL contained lower Ca but higher K in comparison with their corresponding STL. Both company STL had greater Na than their STL and the original tea leaves.

Figure 1 and Figure 2 show the example chromatograms of BTL vs. SBTL samples and GTL vs. SGTL samples as determined by the HPLC examination, respectively. Table 4 shows the mean alkaloid and phenolic contents in different tea leaf products. Caffeine was the major alkaloid in both BTL and GTL. Black tea leaves contained lower amounts of caffeine, alkaloids, and catechins but contained more theaflavins compared with GTL. In GTL, epigallocatechin gallate was the greatest catechin whilst theaflavin-3,3′-digallate was the greatest theaflavin in BTL. The concentrations of most bioactive compounds such as alkaloids, catechins, and theaflavins in each STL were lower than those of the corresponding original tea leaves.

### 3.2. In Vitro Experiments

Table 5 shows the mean values (experiment 1) of rumen in vitro measurements for the primary effect of inclusion levels, tea types, diet types, and their two- and three-way interactions. Across tea and diet types, increasing levels of tea in ruminant diets up to 100 g/kg DM reduced NH_3_, pH, and CH_4_ productions but increased the A:P ratio and had no impact on IVOMD, VFA profiles, tGP, and CO_2_ productions. Black tea, averaged over all the inclusion levels and diet types, had higher NH_3_ and pH but lower A:P ratio and tGP than the green tea inclusions while the other in vitro measurements for both tea types were not different. Across the inclusion levels and tea types, the RH-based diets had higher IVOMD, tGP, CH_4_, and CO_2_ but lower NH_3_ and pH compared with the RS-based diets whilst the other rumen in vitro measurements for both diet types were not different.

Table 6 presents mean values (experiment 2) of rumen in vitro measurements for the main impact of inclusion levels, STL types, diet types, and their two- and three-way interactions. Increasing the levels of STL in ruminant diets from 0 to 100 and 200 g/kg DM, averaged over all the STL and diet types, had no impact on the majority of rumen in vitro measurements. However, this increase reduced both acetate and propionate at 200 g/kg DM and CH_4_ as % total gas at 100 g/kg DM or higher, but it increased A:P ratio at both 100 and 200 g/kg DM. Across the inclusion levels and diet types, there was no difference between black and green STL for most rumen in vitro measurements except that black STL had lower A:P ratio, tGP, and CO_2_ production as L/kg OM than the green STL. The RH-based diet, averaged over all the inclusion levels and STL types, had higher IVOMD, most VFA profiles except iso-Valerate, A:P ratio, tGP, CH_4_, and CO_2_ as L/kg OM but it had lower NH_3_, pH, and CH_4_ output as % total gas in comparison with the RS-based diets. In addition, Figure 3 shows curvilinear (R^2^ = 0.999) patterns of cumulative tGP in the presence of either original tea leaves (left) or STL (right) with either RH- or RS-based diets during rumen in vitro incubation over 24 h. There was a curvilinear effect of increasing hours of incubation on tGP from both original tea leaves and STL during 24 h of their incubation with the RH- and RS-based diets as illustrated by the corresponding predicted equations in Figure 3. 

## 4. Discussion

Based on the chemical characteristics, tea leaves and their STL appeared to be rich in protein, fiber fractions, minerals, and secondary metabolites including tannins and saponins [1] where catechins and theaflavins were the primary tannins in green and black tea products, respectively [3].

Lower NH_3_ concentrations due to tea leaf inclusions could be attributed to their higher secondary metabolite contents such as tannins that can modify the rumen microbial activities. Tannins are able to bind and protect feed proteins from rumen degradation and further reduce NH_3_ output [4]. However, a previous study [6] found that tea tannin inclusion in a diet of feedlot lambs could increase their liveweight gains without reducing feed intakes and protein digestibility. This study confirmed that bound protein was not well-digested in the rumen but was available as a bypass protein that can be absorbed into the small intestine.

Tannins can reduce CH_4_ yield by decreasing the inter-species transfer of H_2_ into methanogenic bacteria and therefore depressing their growth [4,16,36]. Saponins also could reduce CH_4_ production by suppressing the protozoa-related methanogens [9]. In this study, tea leaf inclusions could also decrease pH and this might be related to lower CH_4_ production, as a low ruminal pH can hamper the growth of methanogens as well as protozoa [37]. 

In the rumen, CH_4_ formation is facilitated by the reaction between hydrogen (H_2_) and CO_2_, as shown by the following formula: CO_2_ + 4 H_2_ → CH_4_ + 2 H_2_O, where H_2_ is one of the main end fermentative products by protozoa, fungi, and pure monocultures of several bacteria in the rumen [15,38]. This H_2_ production does not accumulate in the rumen as it is instantaneously utilized by other H_2_-utilizing bacteria such as methanogens *(Methanobrevibacter ruminantium, Methanobacterium formicicum, Methanosarcina mazei, Methanosarcina barkeri, Methanomicrobium mobile)*. The collaboration between H_2_-producing microbes and H_2_-utilizing bacteria is known as “interspecies hydrogen transfer”, where some methanogens are attached to the external pellicles of protozoa. Furthermore, H_2_ along with CO_2_ and other substrates such as formate, acetate, methylamines, dimethyl sulfide, and some alcohols are used by methanogens in the process of forming CH_4_ to produce energy for their own growth. The prevention of accumulating H_2_ is useful for H_2_-producing microbes to further degrade fibrous feed materials as low pressure of H_2_ in the rumen can be maintained [15,16]. As CH_4_ provides no nutritional benefits, its production may represent a dietary energy loss to the animals and damaging for the environment. H_2_ is also produced through acetate and butyrate synthesis during the fermentation of structural carbohydrates as presented in the following equations [16,39]:C_6_H_12_O_6_ + 2H_2_O → 2C_2_H_4_O_2_ (acetate) + 2CO_2_ + 8H
C_6_H_12_O_6_ → C_4_H_8_O_2_ (butyrate) + 2CO_2_ + 4H

Meanwhile, propionate is predominantly formed from non-structural carbohydrate fermentation and acts as a competitive pathway in H_2_ utilization in the rumen, so that its formation is likely to be accompanied by a reduction of CH_4_ production as can be explained by the following equation [15,16,39]: C_6_H_12_O_6_ + 4H → 2C_3_H_6_O_2_ (propionate) + 2H_2_O

Based on the above theory, increased acetate production in the rumen is likely to be followed by higher H_2_ and CH_4_ production, while increased propionate production will be accompanied by lower H_2_ and CH_4_ production. However, decreased CH_4_ production due to tea leaf inclusions in this study was followed by an increased A:P ratio. In this case, the lower CH_4_ concentrations due to tea leaf inclusions might be primarily due to their higher tannin and saponin contents that can lower CH_4_ formation via methanogenesis by inhibiting methanogenic bacteria and protozoa-related methanogens as previously discussed. Another competitive pathway to methanogenesis is reductive acetogenesis that converts H_2_ and CO_2_ into acetate by hydrogenotrophic acetogens, as explained in the following equation [15,17,40]:CO_2_ + 4 H_2_ → CH_4_ + 2 H_2_O (Methanogenesis)
2CO_2_ + 4 H_2_ → CH_3_COOH + 2H_2_O (Acetogenesis).

Under normal circumstances, methanogenesis is likely to be the major pathway for H_2_ utilization in the rumen in comparison with acetogens because of the following three reasons [17,39,40,41]: (1) the conversion of CO_2_ and H_2_ into CH_4_ produces more energy and is thermodynamically more favorable than their conversions to acetate, (2) ruminal acetogens can utilize simple sugars to yield energy so that they seem not to be obligate hydrogenotrophic, and (3) the partial pressure of H_2_ is commonly under the threshold for acetogens even though certain acetogens develop at the thresholds below 1 µmol H_2_/L. However, acetogens can use H_2_ and CO_2_ to form acetate in the rumen when methanogens are inhibited, for example by using 2-bromoethanesulfonic acid [42].

A similar situation occurs in the hindgut fermentation where acetogenesis is more dominant over methanogenesis, resulting in the predominant utilization of H_2_ and CO_2_ by acetogens to form acetate [15,17,43]. Here, acetogenesis seems to be more favorable to compete for the H_2_ utilization by methanogens since acetate produced is absorbed into the blood and utilized as the main source of carbon and energy by animals while CH_4_ is wasted [15]. More acetate production for diets containing tea leaves implies that these tea products could be used as an additive for dairy cattle feeds since elevated acetate availability could increase milk fat synthesis and reduce low-fat milk syndrome [44,45]. 

In comparison with black tea, green tea inclusions had significantly lower rumen NH_3_ and pH but enhanced IVOMD, A:P ratio, and tGP. This suggested that catechins in green tea leaves were perhaps more favorable to protect dietary protein from rumen digestion than theaflavins in black tea leaves. However, relating increased protein binding and decreased rumen NH_3_ concentration with decreased pH due to the green tea inclusions needs further investigation. Lower pH in response to the green tea than black tea inclusions might be due to the faster fermentation of green tea leaves as explained by the greater IVOMD and tGP for the green tea than the black tea leaves. Higher rates of rumen fermentation and increased rate of passage might have resulted in the lower ruminal pH [16,39]. Also, higher IVOMD and tGP for green tea than black tea inclusions showed that green tea was more digested in the in vitro rumen fermentation compared with black tea, which might have contained more resistant components because of the “Maillard browning activities” during the black tea fabricating process. In addition, a greater A:P ratio in green tea than in black tea inclusions meant that green tea was more preferable as a feed additive for dairy cattle to increase milk fat synthesis and reduce low-fat milk syndrome as previously discussed.

Greater IVOMD and tGP for the RH-based diet were expected since RH had greater nutritive value with less fiber content than RS. Again, lower pH for the RH than the RS-based diets might be due to faster fermentation as explained by greater IVOMD and tGP for the RH than the RS-based diets. However, the RH-based diets had significantly higher CH_4_ production as L/kg OM than the RS-based diets, although the RH-based diets had slightly lower CH_4_ concentration as % total gas than the RS-based diets. This suggests that tackling CH_4_ production for example as L/kg OM in ruminants is not only to reduce CH_4_ concentration as % total gas but also to keep tGP at the same level or lower.

Unlike original tea leaves, STL inclusions into ruminant diets could significantly decrease CH_4_ concentration as % total gas but not as L/kg OM. The STL inclusions also did not affect neither NH_3_ concentrations nor pH significantly. This is likely due to there being less secondary metabolites such as tannins and saponins in STL, as the residue after the tea-making process, than their original leaves [1], but STL inclusions could also increase A:P ratio. Similar to the green tea leaves, green STL had higher A:P ratio and faster fermentation as indicated by higher tGP compared with the black STL. Meanwhile, the RH-based diet in the STL experiment had also significantly higher degradability, VFA profiles, A:P ratio, and tGP, confirming that the RH had better nutritive values with more soluble but less fiber content than RS. The RH-based diets also had significantly higher CH_4_ production as L/kg OM but lower CH_4_ concentration as % total gas, confirming that CH_4_ contribution from ruminants was not only affected by CH_4_ concentration as % in the total gas but was also affected by their tGP.

## 5. Conclusions

Tea leaves and their STL are rich in protein, fiber, minerals, caffeine, and polyphenols. It appears from these in vitro studies that tea leaves (especially GTL) and their STL could reduce NH_3_ and CH_4_ production but increase A:P ratio without affecting rumen function and feed degradability. Further animal experiments are required to further examine the potential use of tea leaves and their STL as natural dietary additives for ruminants to improve bypass protein, mitigate CH_4_ production, and reduce low-fat milk syndrome in ruminants at a farm scale.

## Figures and Tables

**Figure 1 animals-12-00305-f001:**
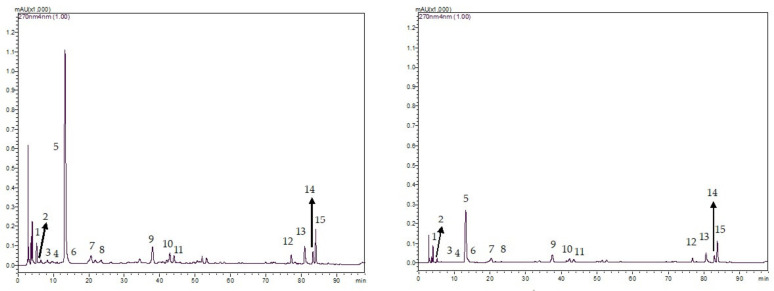
Example HPLC chromatograms of BTL (**left**) vs. SBTL (**right**) samples: (1) Theobromine, (2) gallocatechin, (3) epigallocatechin, (4) catechin, (5) Caffeine, (6) epicatechin, (7) epigallocatechin gallate, (8) gallocatechin gallate, (9) epicatechin gallate, (10) catechin gallate, (11) Rutin, (12) theaflavin, (13) theaflavin-3-gallate, (14) theaflavin-3′-gallate, and (15) theaflavin-3,3′-digallate.

**Figure 2 animals-12-00305-f002:**
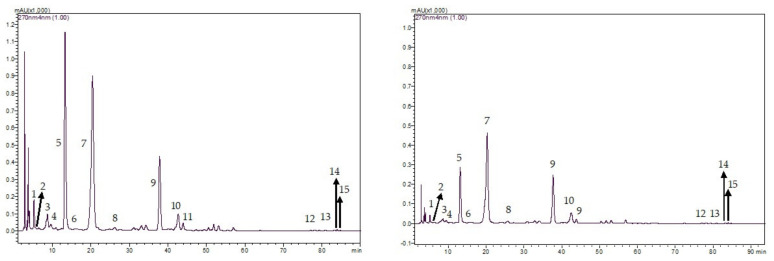
Example HPLC chromatograms of GTL (**left**) vs. SGTL (**right**) samples: (1) Theobromine, (2) gallocatechin, (3) epigallocatechin, (4) catechin, (5) Caffeine, (6) epicatechin, (7) epigallocatechin gallate, (8) gallocatechin gallate, (9) epicatechin gallate, (10) catechin gallate, (11) Rutin, (12) theaflavin, (13) theaflavin-3-gallate, (14) theaflavin-3′-gallate, and (15) theaflavin-3,3′-digallate.

**Figure 3 animals-12-00305-f003:**
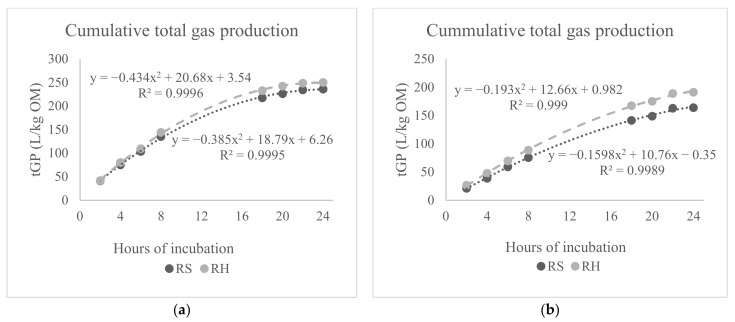
Cumulative total gas production (tGP, L/kg OM) of tea leaves (**a**) and spent tea leaves (STL, (**b**)) in either ryegrass hay (RH, grey) or rice straw (RS, black) based diets during rumen in vitro incubation up to 24 h.

**Table 1 animals-12-00305-t001:** Ingredient compositions (g/kg DM) of the RH- or RS-based experimental diets containing either original tea leaves (BTL, GTL) for the first experiment and their STL (SBTL, SGTL, CSBTL, CSGTL) for the second experiment.

Tea Leaf Products	RH or RS	CON
BTL or GTL		
0	300	700
50	250	700
100	200	700
STL		
0	300	700
100	200	700
200	100	700

BTL, black tea leaves; GTL, green tea leaves; STL, spent tea leaves; SBTL, spent BTL; SGTL, spent GTL; CSBTL, company SBTL; CSGTL, company SGTL; RH, ryegrass hay; RS, rice straw; CON, ruminant concentrate mix.

**Table 2 animals-12-00305-t002:** Mean (*n* = 3) chemical composition (g/kg DM except DM = g DM/kg and ME = MJ/kg DM) of ingredients of the experimental diets with SEM and significance.

Ingredients	DM	OM	Ash	CP	EE	ME	NDFom	ADFom	ADLom	TP	TT	TS
BTL	942 ^a^	939 ^c^	61.4 ^d^	242 ^b^	12.6 ^de^	6.40 ^c^	323 ^f^	309 ^ef^	27.4 ^d^	151 ^b^	133 ^b^	86.1 ^b^
GTL	937 ^a^	938 ^c^	61.8 ^d^	240 ^b^	20.8 ^b^	7.08 ^bc^	254 ^g^	211 ^g^	37.6 ^d^	231 ^a^	204 ^a^	276 ^a^
SBTL	126 ^f^	961 ^a^	38.7 ^f^	234 ^b^	13.5 ^cd^	6.59 ^c^	474 ^d^	410 ^d^	44.5 ^d^	90.9 ^c^	79.1 ^c^	34.0 ^c^
SGTL	134 ^f^	957 ^b^	43.3 ^e^	246 ^a^	23.1 ^b^	7.39 ^b^	405 ^e^	294 ^f^	40.3 ^d^	110 ^b^	99.9 ^b^	55.6 ^b^
CSBTL	205 ^d^	959 ^a^	41.3 ^f^	253 ^a^	12.6 ^de^	6.87 ^bc^	576 ^c^	449 ^c^	48.8 ^d^	34.4 ^de^	31.7 ^d^	12.4 ^d^
CSGTL	170 ^e^	955 ^b^	44.9 ^e^	261 ^a^	17.8 ^bc^	7.49 ^b^	560 ^c^	334 ^e^	42.7 ^d^	44.7 ^d^	39.8 ^d^	26.8 ^cd^
RH	840 ^c^	908 ^e^	92.4 ^b^	200 c	20.2 ^b^	6.79 ^bc^	649 ^b^	507 ^b^	435 ^b^	9.89 ef	2.19 ^e^	16.9 ^cd^
RS	944 ^a^	818 ^e^	182 ^a^	60.4 ^e^	9.9 ^e^	4.01 ^d^	787 ^a^	684 ^a^	598 ^a^	6.12 ^f^	1.08 ^e^	24.3 ^cd^
CON	864 ^b^	921 ^d^	78.9 ^c^	176 ^d^	56.6 ^a^	10.1 ^a^	271 ^g^	144 ^h^	134 ^c^	3.95 ^f^	1.61 ^e^	32.2 ^cd^
SEM	2.37 ***	0.56 ***	0.56 ***	1.77 ***	0.80 ***	0.15 ***	5.35 ***	5.32 ***	4.99 ***	5.74 ***	4.28 ***	5.06 ***

Mean values with different superscripts (^a,b,c,d,e,f,g^) within a column were significantly different at *p* < 0.001 (***); SEM, standard error of mean; *n*, number of replicates; BTL, black tea leaves; GTL, green tea leaves; SBTL, spent BTL; SGTL, spent GTL; CSBTL, company SBTL; CSGTL, company SGTL; RH, ryegrass hay; RS, rice straw; CON, ruminant concentrate mix; DM, dry matter; OM, organic matter; CP, crude protein; ME, metabolizable energy; NDFom, neutral detergent fibre excluding ash; ADFom; acid detergent fibre excluding ash; ADLom, acid detergent lignin excluding ash; TP, total phenols; TT, total tannins; TS, total saponins.

**Table 3 animals-12-00305-t003:** Mean (*n* = 3) mineral components (mg/kg DM) of different tea leaf products with SEM and significance.

Compounds(mg/kg DM)	BTL	GTL	SBTL	SGTL	CSBTL	CSGTL	SEM
Ca	6441 ^c^	6699 ^c^	8281 ^b^	8881 ^b^	10,374 ^a^	10,753 ^a^	171 ***
K	7808 ^a^	8095 ^a^	2521 ^b^	2542 ^b^	632 ^c^	906 ^c^	186 ***
P	2413 ^ab^	2521 ^a^	1904 ^d^	2213 ^bc^	2013 ^cd^	2183 ^bcd^	61.8 ***
Mg	1726 ^bc^	1993 ^a^	1641 ^bc^	1848 ^ab^	1726 ^c^	1864 ^bc^	44.7 ***
Mn	527 ^c^	663 ^b^	536 ^c^	747 ^a^	536 ^c^	804 ^a^	13.6 ***
Fe	116 ^d^	119 ^d^	169 ^bc^	143 ^c^	182 ^d^	346 ^ba^	6.68 ***
Na	150 ^c^	78.2 ^d^	180.1 ^c^	94.6 ^d^	1789 ^a^	1303 ^b^	8.11 ***
Cu	23.8 ^a^	16.9 ^b^	24.0 ^a^	16.6 ^b^	26.9 ^a^	23.8 ^a^	1.16 ***
Zn	21.7 ^ab^	21.2 ^ab^	22.4 ^ab^	19.5 ^b^	23.7 ^a^	20.4 ^b^	0.63 **
Ni	1.69 ^a^	1.58 ^a^	1.17 ^b^	0.49 ^c^	0.69 ^c^	0.40 ^c^	0.08 ***
Cr	1.22 ^b^	1.32 ^b^	1.45 ^b^	1.08 ^b^	1.24 ^b^	2.37 ^a^	0.14 ***
Pb	0.59 ^b^	0.51 ^b^	0.73 ^b^	0.39 ^b^	0.65 ^b^	1.48 ^a^	0.14 **
Cd	0.04	0.04	0.04	0.04	0.07	0.09	0.01 ^NS^

Mean values with different superscripts (^a,b,c,d^) within a row were significantly different at *p* < 0.01 (**) and *p* < 0.001 (***); NS, non-significant; SEM, standard error of mean; *n*, number of replicates; BTL, black tea leaves; GTL, green tea leaves; SBTL, spent BTL; SGTL, spent GTL; CSBTL, company SBTL; CSGTL, company SGTL.

**Table 4 animals-12-00305-t004:** Mean (*n* = 3) alkaloid and phenolic components (g/kg DM) of different tea leaf products with SEM and significance.

Compounds (g/kg DM)	BTL	GTL	SBTL	SGTL	CSBTL	CSGTL	SEM
Theobromine	1.37 ^b^	2.58 ^a^	0.40 ^d^	0.76 ^c^	0.03 ^e^	0.11 ^e^	0.02 ***
Caffeine	27.4 ^b^	28.9 ^a^	9.47 ^c^	9.89 ^c^	0.93 ^d^	0.91 ^d^	0.16 ***
Total alkaloids	28.8 ^b^	31.5 ^a^	9.87 ^d^	10.7 ^c^	0.96 ^e^	1.02 ^e^	0.16 ***
Gallocatechin	n.d.	4.93 ^a^	n.d.	1.61 ^b^	n.d.	0.81 ^c^	0.07 ***
Epigallocatechin	3.51 ^c^	22.4 ^a^	0.80 ^d^	9.02 ^b^	0.07 ^e^	3.22 ^c^	0.14 ***
Catechin	0.40 ^b^	1.30 ^a^	0.14 ^c^	0.40 ^b^	0.03 ^c^	0.14 ^d^	0.01 ***
Epicatechin	0.28 ^c^	2.13 ^a^	0.03 ^d^	1.37 ^b^	0.08 ^d^	0.25 ^c^	0.02 ***
Epigallocatechin gallate	4.45 ^d^	94.6 ^a^	2.30 ^e^	51.8 ^b^	3.69 ^de^	10.7 ^c^	0.41 ***
Gallocatechin gallate	0.60 ^c^	1.15 ^a^	0.16 ^d^	0.85 ^b^	0.16 ^d^	0.75 ^b^	0.03 ***
Epicatechin gallate	5.41 ^c^	25.5 ^a^	2.30 ^e^	14.4 ^b^	1.90 ^e^	4.23 ^d^	0.15 ***
Catechin gallate	1.33 ^c^	3.10 ^a^	0.51 ^e^	1.97 ^b^	0.39 ^e^	0.68 ^d^	0.03 ***
Total catechins	16.0 ^d^	155 ^a^	6.24 ^e^	73.3 ^b^	6.32 ^e^	20.8 ^c^	0.68 ***
Theaflavin	2.33 ^a^	0.28 ^c^	1.44 ^b^	0.18 ^d^	0.33 ^c^	0.07 ^e^	0.01 ***
Theaflavin-3-gallate	4.57 ^a^	0.22 ^d^	3.27 ^b^	0.13 ^de^	0.77 ^c^	0.03 ^e^	0.03 ***
Theaflavin-3′-gallate	2.8 ^a^	0.35 ^d^	2.08 ^b^	0.22 ^e^	0.49 ^c^	0.09 ^f^	0.02 ***
Theaflavin-3,3′-digallate	6.98 ^a^	0.38 ^d^	5.78 ^b^	0.24 ^de^	1.19 ^c^	0.08 ^e^	0.05 ***
Total theaflavins	16.7 ^a^	1.23 ^d^	12.6 ^b^	0.77 ^d^	2.77 ^c^	0.28 ^e^	0.11 ***
Rutin	2.03 ^b^	2.11 ^a^	0.80 ^d^	1.13 ^c^	n.d.	n.d.	0.02 ***

Mean values with different superscripts (^a,b,c,d,e,f^) within a row were significantly different at *p* < 0.001 (***); SEM, standard error of mean; *n*, number of replicates; BTL, black tea leaves; GTL, green tea leaves; SBTL, spent BTL; SGTL, spent GTL; CSBTL, company SBTL; CSGTL, company SGTL.

**Table 5 animals-12-00305-t005:** Mean values of in vitro rumen measurements at 24 h incubation time for the main effect of inclusion levels (L, g/kg DM), tea types (T), diet types (D) and their interactions with SEM and significance.

Measurement	Levels (g/kg DM) *n* = 16			Tea Types *n* = 24		Diet Types *n* = 24		SEM and Significances						
	0	50	100	Black	Green	RH	RS	L	T	D	L*T	L*D	T*D	L*D*T
IVOMD (g/kg OM)	726	715	708	706	727	753	681	7.39 ^NS^	5.74 *	5.89 ***	10.5 ^NS^	11.2 ^NS^	8.33 ^NS^	15.8 ^NS^
NH_3_ (mg/L)	152 ^a^	141 ^b^	126 ^c^	144	135	134	145	1.51 ***	1.24 ***	1.24 ***	2.09 ***	2.25 ^NS^	1.80 ^NS^	3.41 ^NS^
pH	6.69 ^a^	6.67 ^b^	6.67 ^b^	6.68	6.67	6.65	6.70	0.002 ***	0.002 ***	0.002 ***	0.003 **	0.003 ^NS^	0.003 ^NS^	0.005 ^NS^
tVFA (mmol/L)	48.6	48.5	50.2	48.4	49.8	48.6	49.6	1.47 ^NS^	1.18 ^NS^	1.18 ^NS^	2.16 ^NS^	2.16 ^NS^	1.73 ^NS^	3.26 ^NS^
A:P ratio	2.70 ^b^	2.78 ^ab^	2.83 ^b^	2.74	2.81	2.77	2.78	0.03 **	0.02^*^	0.02 ^NS^	0.04 ^NS^	0.04 ^NS^	0.03 ^NS^	0.05 ^NS^
Acetate	28.2	28.3	29.8	28.2	29.4	28.4	29.2	1.04 ^NS^	0.84 ^NS^	0.84 ^NS^	1.52 ^NS^	1.52 ^NS^	1.21 ^NS^	2.30 ^NS^
Propionate	10.4	10.2	10.5	10.3	10.4	10.3	10.5	0.31 ^NS^	0.25 ^NS^	0.25 ^NS^	0.45 ^NS^	0.45 ^NS^	0.36 ^NS^	0.68 ^NS^
iso-Butyrate	0.84	0.83	0.82	0.83	0.83	0.82	0.84	0.01 ^NS^	0.01 ^NS^	0.01 ^NS^	0.02 ^NS^	0.02 ^NS^	0.01 ^NS^	0.03 ^NS^
n-Butyrate	6.70	6.71	6.77	6.69	6.76	6.74	6.71	0.11 ^NS^	0.09 ^NS^	0.09 ^NS^	0.16 ^NS^	0.16 ^NS^	0.13 ^NS^	0.24 ^NS^
iso-Valerate	1.42	1.39	1.35	1.38	1.39	1.38	1.39	0.02 ^NS^	0.01 ^NS^	0.01 ^NS^	0.03 ^NS^	0.03 ^NS^	0.02 ^NS^	0.04 ^NS^
Valerate	1.04	1.02	0.97	1.00	1.02	1.06	0.97	0.01 ^NS^	0.01 ^NS^	0.01 ^NS^	0.02 ^NS^	0.02 ^NS^	0.01 ^NS^	0.03 ^NS^
tGP (L/kg OM)	243	244	243	241	246	251	236	1.40 ^NS^	1.17 **	1.17 ***	2.14 ^NS^	2.14 ^NS^	1.70 ^NS^	3.23 ^NS^
CH_4_ (% total gas)	14.1 ^a^	13.3 ^b^	13.1 ^b^	13.5	13.4	13.3	13.6	0.18 **	0.15 ^NS^	0.15 ^NS^	0.26 ^NS^	0.26 ^NS^	0.21 ^NS^	0.40 ^NS^
CH_4_ (L/kg OM)	34.3 ^a^	32.3 ^b^	31.7 ^b^	32.6	33.0	33.4	32.1	0.45 **	0.37 ^NS^	0.37^*^	0.67 ^NS^	0.67 ^NS^	0.54 ^NS^	1.02 ^NS^
CO_2_ (% total gas)	67.5	66.6	67.4	67.0	67.4	67.6	66.7	1.87 ^NS^	1.49 ^NS^	1.52 ^NS^	2.64 ^NS^	2.82 *	2.21 ^NS^	3.99 ^NS^
CO_2_ (L/kg OM)	165	162	164	162	166	170	158	4.80 ^NS^	3.82 ^NS^	3.92^*^	6.79 ^NS^	6.79 ^NS^	5.67 ^NS^	10.3 ^NS^

Mean values with different superscripts (^a,b,c^) within a row were significantly different at *p* < 0.05 (*), *p* < 0.01 (**), or *p* < 0.001 (***); ^NS^, non-significant; SEM, standard error of mean; *n*, number of replications; RH, ryegrass hay; RS, rice straws; IVOMD, in vitro organic matter degradability; NH_3_, ammonia; tVFA, total volatile fatty acids; A:P ratio, acetate to propionate ratio; tGP, total gas production; CH_4_, methane; CO_2_, carbon dioxide.

**Table 6 animals-12-00305-t006:** Mean values of rumen in vitro measurements at 24 h incubation time for the main effect of inclusion levels (L, g/kg DM), STL types (S), diet types (D) and their interactions with SEM and significance.

Measurement	Levels (g/kg DM) (*n* = 32)			STL Types (*n* = 48)		Diet Types (*n* = 48)		SEM and Significance						
	0	100	200	Black	Green	RH	RS	L	S	D	L*S	L*D	S*D	L*S*D
IVOMD (g/kg OM)	797	799	798	794	803	821	775	6.40 ^NS^	4.18 ^NS^	4.26 ***	9.04 ^NS^	6.13 ^NS^	6.03 ^NS^	13.7 ^NS^
NH_3_ (mg/L)	154	154	153	154	153	152	155	1.05 ^NS^	0.72 ^NS^	0.72 ***	1.49 ^NS^	1.49^*^	1.04 ^NS^	2.11 ^NS^
pH	6.72	6.70	6.69	6.71	6.70	6.67	6.74	0.01 ^NS^	0.01 ^NS^	0.01 ***	0.01 ^NS^	0.01 ^NS^	0.01 ^NS^	0.02 ^NS^
tVFA (mmol/L)	40.7	40.0	38.8	40.4	39.9	42.7	37.6	0.95 ^NS^	0.63 ^NS^	0.63 ***	1.34 ^NS^	1.34 ^NS^	0.89 ^NS^	1.89 ^NS^
A:P ratio	1.75 ^c^	1.83 ^a^	1.81 ^b^	1.79	1.81	1.84	1.76	0.005 ***	0.005 **	0.005 ***	0.010 *	0.010 ***	0.006 ^NS^	0.014 ^NS^
Acetate	22.2 ^a^	21.7 ^ab^	21.3 ^b^	22.1	22.0	23.6	20.5	0.51 *	0.34 ^NS^	0.34 ***	0.73 ^NS^	0.73 ^NS^	0.49 ^NS^	1.03 ^NS^
Propionate	12.7 ^a^	12.4 ^ab^	11.8 ^b^	12.4	12.2	12.9	11.7	0.29 *	0.19 ^NS^	0.19 ***	0.40 ^NS^	0.40 ^NS^	0.27 ^NS^	0.57 ^NS^
iso-Butyrate	0.56	0.55	0.53	0.58	0.51	0.58	0.51	0.02 ^NS^	0.01 ^NS^	0.01 ***	0.02 ^NS^	0.02 ^NS^	0.02 ^NS^	0.03 ^NS^
n-Butyrate	3.44	3.53	3.35	3.48	3.40	3.66	3.23	0.09 ^NS^	0.06 ^NS^	0.06 ***	0.12 ^NS^	0.12 ^NS^	0.08 ^NS^	0.18 ^NS^
iso-Valerate	0.69	0.71	0.68	0.71	0.68	0.73	0.65	0.02 ^NS^	0.02 ^NS^	0.02 **	0.03 ^NS^	0.03 ^NS^	0.02 ^NS^	0.05 ^NS^
Valerate	1.15	1.14	1.09	1.15	1.10	1.28	0.98	0.03 ^NS^	0.02 ^NS^	0.02 **	0.04 ^NS^	0.04 ^NS^	0.03 ^NS^	0.06 ^NS^
tGP (L/kg OM)	175	179	177	175	179	193	161	1.55 ^NS^	1.04 **	1.05 ***	2.19 ^NS^	2.19 ***	1.49 ^NS^	3.10 ^NS^
CH_4_ (% total gas)	13.8 ^a^	13.3 ^b^	13.4 ^b^	13.6	13.5	13.3	13.7	0.12 **	0.08 ^NS^	0.08 **	0.16 ^NS^	0.16 ^NS^	0.11 ^NS^	0.23 ^NS^
CH_4_ (L/kg OM)	24.2	23.8	23.6	23.7	24.1	25.7	22.0	0.29 ^NS^	0.20 ^NS^	0.20 **	0.41 ^NS^	0.41 **	0.28 ^NS^	0.58 ^NS^
CO_2_ (% total gas)	68.7	67.1	65.8	66.7	67.7	70.1	64.3	1.22 ^NS^	0.83 ^NS^	0.83 ***	1.72 ^NS^	1.72 ^NS^	1.17 ^NS^	2.44 ^NS^
CO_2_ (L/kg OM)	121	121	117	117	122	135	104	2.46 ^NS^	1.68 *	1.68 ***	3.48 ^NS^	3.48 **	2.40 ^NS^	4.92^NS^

Mean values with different superscripts (^a,b,c^) within a row were significantly different at *p* < 0.05 (*), *p* < 0.01 (**), or *p* < 0.001 (***); ^NS^, non-significant; SEM, standard error of mean; *n*, number of replications; STL, spent tea leaves; RH, ryegrass hay; RS, rice straws; IVOMD, in vitro organic matter degradability; NH3, ammonia; tVFA, total volatile fatty acids; A:P ratio, acetate to propionate ratio; tGP, total gas production; CH_4_, methane; CO_2_, carbon dioxide.

## Data Availability

The data supporting the reported results in this study are available on request from the corresponding author.

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
