# Peer review of "Biochemical Properties of Black and Green Teas and Their Insoluble Residues as Natural Dietary Additives to Optimize In Vitro Rumen Degradability and Fermentation but Reduce Methane in Sheep"

_animals, 2022, doi:10.3390/ani12030305_

Round 1
Reviewer 1 Report
The paper if of scientific interest and fits within the scope of Animals. However it should be greatly improved before being suitable for publication.
The authors propose the use of tea leaves in ruminant nutrition at very high concentrations (up to 20% DM). I wonder if it is economically profitable, and consequently the paper has a practical interest.
The results section is very poorly written. The sentence “with SEM and P values indicating the level of significance” is repeated for all parameters across the section and should be avoided. Moreover, it is not needed to add the formula “UNIT, P value” (i.e. (g/kg DM, P < 0.001) for all parameters; this information is given in the Tables.
It is not clear why some parameters, as minerals concentrations, have been determined since no discussion in made about them; it seems that Cu concentrations are quite high which could be a problem for ruminants, particularly for sheep when given at high concentrations in the diet. The results of the patterns of cumulative tGP given in Figure 3 are neither described or discussed.
Most of the discussion is just a theoretical description of ruminal metabolism no connected with the results of the paper and should be deleted. The authors should focus on their results avoiding interpretation/speculations not inferred from the results
Reviewer 2 Report
Dear authors,
Although there are numerous studies investigating the in vitro impact of tannins, tea leaves, and polyphenolic compounds in rumen fermentation parameters, and I would also expect to find some microbiome analysis in the present study, I think that your manuscript is well written and deserve to be published.
Please find a few comments below:
1) The ruminant species should be mentioned in the title.
2) In my opinion, it is unsustainable to supplement high genetic merit ruminants with 10-20% tea leaves or STL considering the food-feed competition. It may bring positive outcomes in an in vitro experiment but it does not make any sense on a farm scale.
3) More information about donor lambs is crucial. Did they consume a constant diet before? Age of the lambs? Was the rumen microbiome mature? physically or artificially reared? This information is closely linked with rumen biochemistry.
4) Figures 1 and 2 are of little help without the corresponding compounds in each peak. They could be moved as supplementary material.
Table 4. Please don't use abbreviations in the polyphenols. It is hard to be followed.
5) I would avoid discussing about lower abundance of methanogens and protozoa since you did not investigate them in the present study, and their abundance is not always positively correlated with their metabolic outcomes.
6) Please use italics in methanogenic species.
Reviewer 3 Report
Dear Authors,
the present manuscript ("Biochemical Properties of Black and Green Teas and Their Insoluble Residues as Natural Dietary Additives to Optimize in vitro Rumen Degradability and Fermentation but Reduce Methane Emission") is important and actual work, but it is not in a frame of the major journal scopes. The main problem are the following : no experiments using animals and the majority of the experiments have made (in vitro) without reasonable link to animals. I do not doubt the technical quality of the work, but feel that there is an insufficient impact on a broader readership. It will be very useful, if the authors can provide some experiments using animals or discussions of similar experiments.

Round 2
Reviewer 3 Report
the present manuscript ("Biochemical Properties of Black and Green Teas and Their Insoluble Residues as Natural Dietary Additives to Optimize in vitro Rumen Degradability and Fermentation but Reduce Methane Emission") is important and actual work. The authors corrected the main problems in description and some weak points in this manuscript (in the parts “2. Materials and Methods” and “3.2. In vitro experiments”, as well as in the tables and Figures 1 and 2). I do not doubt the technical quality of the work and feel that there is a sufficient impact on a broader readership. The present manuscript can be accepted after minor revision (text editing, i.e. English language and style are minor spell check required).

Author Response
First of all, we would like to thank Reviewer 3 for supportive feedback and appreciation. We checked the manuscript carefully for English language, style, and spelling to include references and we revised any mistakes. Revisions can be seen in the manuscript.
